# Olaparib not cost-effective as maintenance therapy for platinum-sensitive, *BRCA1/2* germline-mutated metastatic pancreatic cancer

**Tarun Mehra**[1☯]*, **Judith E. Lupatsch**[2☯], **Thibaud Kössler**[3], **Konstantin Dedes**[4], **Alexander Reinhard Siebenhüner**[5], **Roger von Moos**[6], **Andreas Wicki**[7‡], **Matthias E. Schwenkglenks**[8‡]

1 Department of Medical Oncology and Hematology, University Hospital of Zurich, Zurich, Switzerland,
2 Department of Public Health, Institute of Pharmaceutical Medicine (ECPM) and Health Economics Facility, University of Basel, Basel, Switzerland, 3 Service d'oncologie, Hôpitaux Universitaires Genève, Genève, Switzerland, 4 Breast Cancer Center Zürichsee, Zurich, Switzerland, 5 Onkozentrum Hirslanden Zurich AG and University of Zurich, Zurich, Switzerland, 6 Department of Medical Oncology and Hematology, Cantonal Hospital of Graubünden, Chur, Switzerland, 7 Department of Medical Oncology and Hematology, University Hospital of Zurich and University of Zurich, Zurich, Switzerland, 8 Department of Public Health and Head of Research, Health Economics Facility, Institute of Pharmaceutical Medicine (ECPM), University of Basel, Basel, Switzerland

☯ These authors contributed equally to this work.
‡ AW and MES also contributed equally to this work.
* tarun.mehra@usz.ch

**Data Availability Statement:** We used pre-published open-access data from the POLO trial. All additional data (cost, outcome and

## Abstract

### Objective

To assess the cost-effectiveness and budget impact of olaparib as a maintenance therapy in platinum-responsive, metastatic pancreatic cancer patients harboring a germline *BRCA1/2* mutation, using the Swiss context as a model.

### Methods

Based on data from the POLO trial, published literature and local cost data, we developed a partitioned survival model of olaparib maintenance including full costs for *BRCA1/2* germline testing compared to FOLFIRI maintenance chemotherapy and watch-and-wait. We calculated the incremental cost-effectiveness ratio (ICER) for the base case and several scenario analyses and estimated 5-year budget impact.

### Results

Comparing olaparib with watch-and wait and maintenance chemotherapy resulted in incremental cost-effectiveness ratios of CHF 2,711,716 and CHF 2,217,083 per QALY gained, respectively. The 5-year costs for the olaparib strategy in Switzerland would be CHF 22.4 million, of which CHF 11.4 million would be accounted for by germline *BRCA1/2* screening of the potentially eligible population. This would amount to a budget impact of CHF 15.4 million (USD 16.9 million) versus watch-and-wait.

epidemiological data) are publihed in the supplementary file.

**Funding:** This study was partially funded by the Swiss Study Group for Clinical Cancer Research (SAKK, Schweizerische Arbeitsgemeinschaft für Klinische Krebsforschung). The funder had no role in study design, data collection and analysis, decision to publish, or preparation of the manuscript. There was no additional external funding received for this study".

**Competing interests:** All authors have read the journal's policy and the authors of this manuscript have the following competing interests. AW: no conflicts of interest for this article AS: no conflicts of interest for this article KD: no conflicts of interest for this article TK: no conflicts of interest for this article JL: no conflicts of interest for this article MS: unrelated to the present work, research funding from AbbVie, Biogen, Bristol Myers Squibb, Merck Sharpe & Dohme, Mundipharma, Novartis, and Roche via employment institution; personal fees from Sandoz RvM: consultancy fees from Astra Zeneca, Eili Lilly, Gilead Science, GlaxoSmithKline, Merck, MSD, Novartis, Pierre Fabre, Pharmamar, Sanofi and Vifor. Travel grants from Pierre Fabre, Takeda TM: no conflicts of interest for this article This study was partially funded by Swiss Study Group for Clinical Oncological Research (SAKK, Schweizerische Arbeitsgemeinschaft für Klinische Krebsforschung). This does not alter our adherence to PLOS ONE policies on sharing data and materials.

## Conclusions

Olaparib is not a cost-effective maintenance treatment option. Companion diagnostics are an equally important cost driver as the drug itself.

## Introduction

Pancreatic cancer accounts for 3% of all cancer diagnoses in the United States and for 7% of cancer deaths [1]. Over half the cases are diagnosed at an advanced stage which has a 5-year overall survival rate of less than 10% [2]. Treatment options for advanced disease are limited. In the PRODIGE4/ACCORD11 trial, platinum-based chemotherapy with FOLFIRINOX demonstrated the best results so far in this setting. Median overall survival (OS) was 11.1 months with FOLFIRINOX compared to 6.8 months with gemcitabine, albeit at the price of substantially higher toxicity [3]. FOLFIRINOX has been the first-line standard of care since, along with gemcitabine and nab-paclitaxel [4], with very limited therapy options thereafter.

Olaparib is a poly-ADP-ribose-polymerase (PARP) inhibitor, which acts by impeding homologous recombination repair (HRR) of damaged DNA strands. In patients with constitutionally impaired HRR mechanisms due to germline mutations of genes such as *BRCA1* or *BRCA2*, further inhibition of HRR leads to cell death. Germline *BRCA1/2* (*gBRCA1/2*) mutational prevalence in patients with pancreatic cancer ranges from 3.8% to 7%, with the higher value probably due to a higher percentage of patients with Ashkenazi descendency in the examined populations [5–7]. The POLO trial examined the effect of olaparib maintenance therapy in *BRCA1/2*-germline-mutated patients with metastatic pancreatic adenocarcinoma that had not progressed under first-line platinum-based chemotherapy. The trial reported a significant increase in PFS, a median of 7.4 months in the treatment group versus 3.8 months in the placebo group [7]. However, the trial showed no overall survival or quality of life improvement [7–9]. Antitumor activity of olaparib in advanced pancreatic cancer has been confirmed in a real-life setting via the open-label phase II TAPUR basket trial [10].

Reporting an incremental cost-effectiveness-ratio (ICER) of United States dollars (USD) 265,290 per additional quality-adjusted life-year (QALY) gained, previous results for the USA questioned the cost-effectiveness of olaparib maintenance therapy. Only when the ICER calculation was restricted to quality-adjusted progression-free survival time, a value just below the USD 200,000 threshold was reached [11]. Given the lack of overall survival or quality-of-life benefit in the POLO trial, the interpretation of this result remains problematic. Furthermore, feasibility of testing all eligible pancreatic cancer patients for *gBRCA1/2* mutations and implications for healthcare systems remain unclear. The present work aimed to (i) estimate the cost-effectiveness of olaparib maintenance therapy for eligible patients with metastatic *gBRCA1/2*-mutated adenocarcinoma of the pancreas in Switzerland and to (ii) measure the impact on the total health care budget if all potentially eligible patients were treated.

## Methods

A list of abbreviations can be found in S1 File (S1 Table in S1 File).

We developed a partitioned survival model (S1 Methods in S1 File) comprising three health states representing stable disease pre-progression, progressive disease and death. The model compared an olaparib maintenance strategy with two standard of care comparator strategies involving maintenance chemotherapy with FOLFIRI (representing a possible clinical

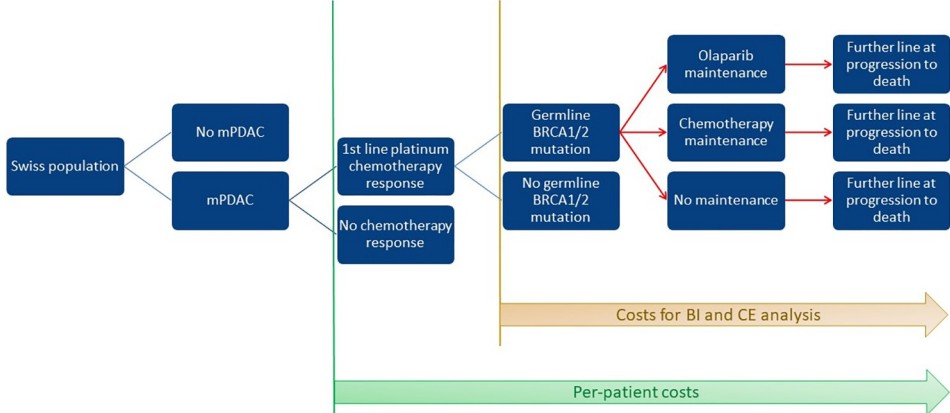

**Fig 1. Clinical pathway model.** mPDAC: metastatic pancreatic ductal adeno-carcinoma. 1st line platinum chemotherapy: FOLFIRINOX. Chemotherapy maintenance: FOLFIRI. Further line chemotherapy: gemcitabine & nab-paclitaxel. All pathways assume 1.5 months of palliative end-of-life care.

standard) and a watch-and-wait approach (essentially mimicking the comparator arm in the POLO trial). With a focus on practical relevance, the latter was regarded as the main comparator strategy for which a full set of uncertainty analyses was performed. Fig 1 is a graphical description of the treatment strategies (Fig 1). The olaparib maintenance strategy included costs for *gBRCA1/2* testing of all pancreatic cancer patients evaluated for eligibility. Outcomes included quality-adjusted life-years (QALYs), costs in total and by category, and incremental cost-effectiveness ratios. The time horizon was 10 years, essentially implying life-long for the patient population under study. Costs and effects were discounted by 3% per year in the base case and evaluated against a hypothetical willingness-to-pay (WTP) threshold of Swiss francs (CHF) 100,000 per QALY gained. All costs were assessed from the Swiss statutory health insurance perspective. We calculated incremental cost-effectiveness ratios (ICER) comparing all three strategies.

An additional budget impact model was based on estimates of how many patients in Switzerland could potentially receive olaparib over 5 years, and undiscounted results from the cost-effectiveness model for years 1 to 5. We assumed here that only patients with a newly established potential eligibility for olaparib would be considered, excluding a switching of treatment protocols for patients already potentially eligible before year 1 of the budget impact model.

Model inputs representing the PFS and OS of patients, and adverse event probabilities, were based on the POLO trial [7]. As the trial did not include a chemotherapy maintenance arm, PFS and OS from the placebo arm were assumed to also reflect survival in the chemotherapy maintenance strategy. As the olaparib OS curve in the POLO trial does not differ significantly from the placebo OS curve, only the placebo OS curve was assumed to apply for all three strategies, in the base case. Kaplan-Meier plots were digitalized and an underlying time-to-event dataset estimated, as a basis for fitting suitable survival distributions.

We used the term disease control rate to denote the proportion of patients without evidence of disease from progression after first line therapy at randomization for a maintenance strategy, as in the POLO trial. Utility parameters, representing health-related quality of life on a scale from 0 (death) to 1 (perfect health) were based on published literature (Table 1). Medical resource use assumptions were based on authors' experience with Swiss clinical practice and unit cost parameters were drawn from publicly available sources, as detailed in S2 Table in S1 File. Frequency of adverse events were retrieved from the literature [7, 12, 13]. In order to

**Table 1. Utilities.**

| Utilities | Value | Sensitivity | Reference |
|---|---|---|---|
| *Base case* | | | |
| PFS | 0.81 | SD 0.15 | [29] |
| PD | 0.73 | SD 0.18 | [29] |
| *Scenario* | | | |
| PD | 0.58 | | [15] |

PFS = progression free survival, PD = progressive disease

estimate the population of patients potentially eligible for olaparib treatment, Swiss demographic and cancer registry data were combined with epidemiological information drawn from international sources.

The methods are described in detail in the Supplementary Methods section of the Supplementary Files (S3-S5 Tables, S1, S2 Figs in S1 File).

As we performed a cost-effectiveness analysis based on published data and as the additional data which was collected was retrieved from public domain, no patients were recruited or primary patient data retrospectively analyzed. Therefore, no ethics approval was required.

The manuscript conforms to the Consolidated Health Economic Evaluation Reports Standards (CHEERS) statement [14].

## Results

### Cost-effectiveness analysis

Of the three strategies, watch-and-wait was the least expensive with total expected per-patient costs of CHF 110,445 (USD 121,490), followed by the FOLFIRI maintenance strategy with CHF 136,252 (USD 149,877) and the olaparib strategy with CHF 251,657 (USD 276,823) (including *BRCA1/2* testing of all potentially eligible patients). The watch-and-wait and maintenance strategies both yielded 1.43 QALYs and the olaparib strategy 1.48. The maintenance strategy was absolutely dominated by the watch-and-wait strategy, given the same amount of QALYs but higher costs. Comparing olaparib with watch-and-wait resulted in an ICER of CHF 2,711,716 (USD 2,982,888) per QALY gained. Comparing FOLFIRI-maintenance with olaparib resulted in an ICER of CHF 2,217,083 (USD 2,438) per QALY gained. Without testing the whole potentially eligible cohort for gBRCA1/2 mutations, the last-mentioned ICERs would change substantially to CHF 814,167 (USD 895,583) and CHF 3128,732 (USD 350,605) per QALY gained, respectively.

Substantially lower ICERs were seen in some scenario analyses: a price reduction of 25% for olaparib would yield an ICER of CHF 2,322,345 (USD 2,554,580) per QALY gained versus watch-and-wait; using a less conservative modelling approach by modelling OS in the olaparib strategy on the basis of the observed olaparib OS survival curve resulted in an ICER of CHF 473,083 (USD 520,392). Using the utility for progressive disease (PD) from Gharaibeh et al. [15] lead to an ICER of CHF 943,604 (USD 1,037,964); additional combination with the olaparib OS curve reduced it further to CHF 283,188 (USD 276,823). Adding extra disutility of adverse events did not lead to substantial changes (ICER of CHF 1,916,845 or USD 2,108,529) (Table 2). In most scenario analyses, QALY results were similar as in the base case analysis; using the olaparib OS course yielded the highest observed QALY value of 1.79.

The univariate sensitivity analyses showed that the costs of gBRCA1/2 testing and olaparib, and utility values, had the largest influence on cost-effectiveness (S3 Fig in S1 File). The PSA

**Table 2. Results from cost-effectivess analysis.**

| Strategy | Cost | Incr Cost | Eff | Incr Eff | ICER[3] | ICER without BRCA cohort testing |
|---|---|---|---|---|---|---|
| **Base Case** | | | | | | |
| Observation | 110,445 | 141,212 | 1.43 | 0.05 | 2,711,716 | 814,167 |
| Maintenance | 136,252 | 115,405 | 1.43 | 0.05 | 2,217,083 | 318,732 |
| Olaparib | 251,657 | | 1.48 | | | |
| **Scenarios** | | | | | | |
| 1 Olaparib—25% | 231,330 | 120,884 | 1.48 | 0.05 | 2,322,345 | 432,815 |
| 2 Olaparib OS curve | 279,133 | 168,688 | 1.79 | 0.36 | 468,605 | 195,491 |
| 3 Different PD QALY[1] | | | | | | |
| Observation | 110,445 | | 1.22 | | | |
| Olaparib | 251,657 | 141,212 | 1.37 | 0.15 | 943,604 | 283,188 |
| 4 Different PD QALY[1] + Olap OS curve | | | | | | |
| Olaparib | 279,133 | 168,688 | 1.61 | 0.39 | 429,973 | 195,492 |
| 5 Disutility for Adverse Events[2] | | | | | | |
| Observation | 110,445 | | 1.08 | | | |
| Olaparib | 251,657 | 141,212 | 1.16 | 0.07 | 1,916,845 | 575,514 |
| 6 BRCA test USD 200 (180 CHF) | | | | | | |
| Olaparib | 153,043 | 42,598 | 1.48 | 0.05 | 818,007 | |

gBRCA1/2: germline BRCA1 and BRCA2. All costs in CHF.

[1] Progressive disease utility: 0.58 [15], instead of 0.73 [29]

[2] Disutility of– 0.2

[3] Considering gBRCA1/2 testing costs of all eligible patients

The olaparib maintenance strategy included costs for gBRCA1/2 testing of all pancreatic cancer patients evaluated for eligibility.

indicated the base case results to be robust. None of the 1,000 iterations comparing the olaparib and watch-and-wait strategies yielded results in regions where cost-effectiveness could be regarded as given (S4 Fig in S1 File).

## Budget-impact analysis

We estimated an incidence of 579 patients per year in Switzerland with platinum-sensitive metastatic pancreatic adenocarcinoma [16] and thus eligible for gBRCA1/2 screening (S2 Table in S1 File). Postulating a gBRCA1/2 mutation prevalence of 4%, we calculated a target population of 23 new cases per year eligible for olaparib maintenance therapy, totalling to 115 over the 5-year period.

The 5-year costs for the olaparib maintenance strategy would be CHF 22.4 million (USD 24.6 million), for FOLFIRI maintenance CHF 9.7 million (USD 10.6 million), and watch-and-wait CHF 7.0 million (USD 7.7 million) (Table 3), accounting for differences in the attrition rate between the three strategies and increased costs for the treatment of side-effects in the FOLFIRI maintenance strategy. In the olaparib strategy, the gBRCA1/2 screening costs would amount to CHF 11.4 million (USD 12.5 million) over 5 years. Results are depicted in the (S5 Fig in S1 File). To summarize, the 5-year budget impact for the olaparib strategy versus watch and wait would amount to CHF 15.4 million (USD 16.9 million) and the 5-year budget impact for the FOLFIRI maintenance strategy versus watch and wait CHF 2.6 million (USD 2.9 million).

In our scenario analyses, the largest increase in the 5-year cost estimate for the olaparib strategy would occur if the incidence of gBRCA1/2 mutations was 7% (cost increase of 39%)

**Table 3. Base case and scenario analyses of 5-year total costs, for the treatment of platinum-sensitive germline *BRCA 1/2* mutated pancreatic cancer in Switzerland with either olaparib maintenance, FOLFIRI maintenance or watch and wait (yearly incidence of n = 23).**

| Scenario | N | Olaparib | FOLFIRI | Watch & Wait |
|---|---|---|---|---|
| Base Case | 115 | 22'403'244 | 9'655'132 | 7'006'720 |
| BRCA 2% | 60 | 17'170'354 | 5'080'834 | 3'696'292 |
| BRCA 7% | 205 | 31'116'587 | 17'279'394 | 12'554'029 |
| DCR 60% | 100 | 19'333'012 | 8'376'468 | 6'074'589 |
| DCR 50% | 85 | 16'264'845 | 7'117'727 | 5'161'444 |
| NGS germline 200 CHF | 115 | 13'344'789 | 9'655'132 | 7'006'720 |
| Olaparib 25% cost reduction | 115 | 20'831'527 | 9'655'132 | 7'006'720 |

DCR: disease control rate; N: size of the target population for the estimated period of 5 years, i.e. number of cases in Switzerland with metastatic ductal adenocarcinoma of the pancreas with gBRCA1/2 mutation and controlled disease after 1st line platinum-based chemotherapy in the 5-year period, accounting for attrition; NGS: next generation sequencing. All costs in Swiss francs (CHF). Costs include maintenance therapy, 2nd line chemotherapy with gemcitabine and nab-paclitaxel, as well as palliative end of life care, including costs of side-effects, routine medical visits and laboratory testing as well as imaging costs. The olaparib maintenance strategy included costs for gBRCA1/2 testing of all pancreatic cancer patients evaluated for eligibility. Costs for standard care first line platinum-based chemotherapy and routine somatic genetic testing are not included.

NGS germline 200 CHF: NGS costs for germline sequencing reduced from 3,329 CHF to 200 CHF; BRCA 2%: incidence of *BRCA 1/2* germline mutation prevalence decreased from 4% to 2%; BRCA 7%: incidence of *BRCA 1/2* germline mutation prevalence increased from 4% to 7%; DCR 60%: DCR reduced from 70% to 60%; DCR 50%: DCR reduced from 70% to 50%; Olaparib cost reduction: drug price reduction of olaparib by 25%.

and the largest cost reduction would be seen if the cost for *gBRCA1/2* testing fell to CHF 200 (-40%). A reduction in the price of olaparib bei 25% would reduce the total 5-year cost by 7%. (S6 Fig in S1 File and Table 3). A comparison of the costs for the whole per-patient treatment pathway per scenario can be found in the supplementary file (S6 and S7 Tables in S1 File).

# Discussion

## Cost-effectiveness

Using a modelling approch including the entire expected costs of olaparib treatment in patients with metastatic pancreatic cancer led to ICERs of olaparib which were far from cost-effective when considering standard willingness-to-pay thresholds. Including total costs for *gBRCA1/2*-testing of all potentially eligible patients and assuming no overall survival differences between the olaparib and watch-and-wait strategies led to an ICER of around CHF 2,700,000 per QALY gained. This result was driven by substantially higher costs of the olaparib strategy and, on the effectiveness side, by an only small gain in quality-adjusted lifetime, emerging from a longer time lived without progression and shorter time lived with progression. Small changes in the analysis strategy impacted the ICER values rather substantially as our scenario analyses demonstrated. However, ICERs usually regarded as cost-effective were not reached. Even when simulating considerably lower costs for *gBRCA1/2* testing of of less than tenfold the Swiss market price, leading to an decrease of the incremental cost in comparison to the watch-and-wait strategy by two-thirds, the cost-per-QALY gained still amounted to over USD 800,000 (Table 2 scenario 6).

Previous cost-effectiveness analyses do not seem to have considered the costs of routine molecular profiling, or *gBRCA1/2* testing. Using the olaparib OS curve and a lower utility for PD than in our base case (0.58 [15] instead of 0.73), and thus a combination of utility results from different sources, an analysis for the US healthcare system estimated an ICER of olaparib compared to placebo (as in our watch-and-wait strategy) of USD 265,290 (CHF 260, 301) per QALY gained [11]. The authors assumed higher per-day treatment costs for olaparib of USD

244.52 (approximately CHF 222.30), in accordance with higher US drug prices, in comparison to our assumption of 191.4 CHF per day. Mimicking this analysis in scenario 4 based on our model, but maintaining our Swiss cost assumptions, we estimated an even smaller ICER, of CHF 179,264 (USD 197,190). However, we regarded it as important for our base case to extract utilities for PFS and PD from the same publication as combining utilities observed in different population may lead to inconsistencies. Since the POLO trial did not show a survival gain from olaparib, we also assumed no such gain in our base case. An analyis from a Chinese perspective concluded that at a WTP threshhold of USD 28,256 and an ICER of USD 34,122 per QALY gained, olaparib was not cost-effective [17] The same conclusion was reached by a recently published Canadian study, which calculated an ICER of CAD 329,517 (USD 245,826) per QALY at a WTP threshold of CAD 50,000 (USD 37,301) [18]. A further study came to opposite conclusions, with calculated ICERs of USD 6,694 and USD 13,327 per QALY gained, at WTP thresholds of USD 30,829 and USD 50,000 per QALY gained, for China and the USA, respectively [19]. However, this latter study calculated a difference of 8.77 QALYs per patient for the olaparib strategy in comparison to the placebo strategy, at a cost difference of USD 116,881. In light of the absence of an OS benefit or differences in QoL in the POLO trial [8, 9], we question these results.

## Budget impact

We calculated a cost of CHF 22.4 million for the olaparib strategy, over 5 years and for a total of 115 eligible patients. *gBRCA1/2* screening was responsible for 51% of these costs. The 5-year cost of the chemotherapy maintenance and watch-and-wait strategies were 43% and 31%, respectively, of the cost of the olaparib strategy. In perspective, expenditures for total cancer care in Switzerland were CHF 4.366 bn (USD 4.803 bn) in 2018, based on an estimated prevalence of 130,500 cancer patients [20], of which CHF 801 million (USD 881 million) were spent on drugs [21].

Our results make understandable the hesitancy of countries to adopt reimbursement for PARP inhibitors. For example, Ireland did not recommend olaparib for reimbursement as a maintenance treatment of platinum-sensitive, high-grade epithelial ovarian, fallopian tube or primary peritoneal cancer, citing average treatment costs per patient of approximately USD 200,000 and a budget impact of USD 27.8 million over 5 years for expected total drug costs for an intention-to-treat population of 100–150 patients [22], despite a proven OS benefit. To put these estimated into perspective, Ireland has a population of 4.9 million and an annual health care budget of approximately USD 27.5 billion [23].

## Strengths and limitations

Our analysis has several strenghts: (i) cost completeness and transparency: we were able to include the full medical management costs beginning with molecular profiling of all patients potentially eligible for olaparib treatment. To our knowledge, our study is the first to consider the screening cost of the entire potentially eligibly population in a cost-effectiveness analysis of olaparib. (ii) Inclusion of upstream diagnostic costs: we included the costs of companion diagnostics required to screen the entire potentially eligebele population. Including these costs, we estimated the ICER per QALY gained based on overall survival, observed and then extrapolated during 10 years, as our main result that we validated against the expected survival of the patient population. The robustness of the assumptions was tested in several scenario analyses. (iii) We included a budget impact analysis in which we considered the expected patient population for Switzerland. This analysis stresses that there are only very few patients eligible for olaparib treatment. However, as Switzerland and the USA are socioeconomically comparable,

we assume that our cost-effectiveness and budget impact results can be extrapolated to a US setting. Furthermore, to our knowledge, our study is the first to highlight the substantial financial burden of companion diagnostic testing on the budget impact of introducing olaparib in metastatic pancreatic cancer; a finding which should be considered when evaluating other targeted therapies.

There are also some limitations of our approach. As our results rely on a single trial only, the robustness and external validity may be questioned in the same way as that of the clinical trial results. In addition, We had to extrapolate overall survival; yet, our extrapolation yielded 10-year survival results close to the those observed in the SEER database [24]. We do not expect our results to change substantially if the recently published mature survival data [8] from the POLO trial were incorporated into our analysis. Furthermore, the analysis is highly depended on the utility parameters used. We could not differentiate between utilities for the olaparib versus other strategies; however, in a scenario analysis we subtracted a disutility for treatment-specific AE occurrences, which did not alter the results substantially. Notably, germline *BRCA* testing of pancreatic cancer patients may not only benefit the patients directly, but family members as well, with a negative result reducing anxiety and a positive result enabling targeted preventive measures. However, these effects are hard to quantify and were hence not considered in our analysis. Additionally, costs for *gBRCA1/2* screening may vary substantially. Indeed, costs of molecular profiling vary substantially between countries. In comparison to the per-patient cost of CHF 3,329 for *gBRCA1/2* screening in Switzerland, costs in Australia amount to 1,200 Australian Dollars (CHF 757.51), which is less than one quarter of the Swiss cost [25]. In the USA, USD 2,760 are billed to Medicare/Medicaid, for *BRCA1/2* testing based on the most frequent test, offered by Myriad, which amounts to 80% of the cost in Switzerland [26]. Furthermore, we assumed that all patients would receive FOLFIRINOX in the first line of treatment, as non-progression under platinum-based chemotherapy is a prerequisite for considering olaparib eligibility. However, in our clinical experience, far from all patients are considered eligible for FOLFIRINOX triplet therapy. Many receive a doublet therapy such as gemcitabine and nab-paclitaxel in the first line. Hence, the proportion of patients potentially eligible for olaparib treatment may be lower and the budget impact of olaparib smaller. We acknowledge that the considered timeframe of 5 years is long for the population considered due to the high attrition rate as well as the shorter 3-year time frame published for the survival data of the POLO trial. However, we decided to maintain the 5-year horizon as it is a standardized observation period.

## Interpretation and outlook

Our results show that individual, per-patient treatment costs for patients with metastatic, *gBRCA1/2* mutated, platinum-sensitive pancreatic cancer are driven by the high costs of olaparib, whereas overall cost-effectiveness and budget impact is driven by the costs of companion diagnostics.

A benefit in PFS in the absence of an OS benefit is only relevant from a patient's perspective if accompanied by a meaningful improvement in QoL or substantially less toxicity. Indeed, this is reflected by widely accepted value frameforks for interpreting clinical benefit of oncological treatments, namely the ASCO Value Framework [27] and the ESMO Magnitude of Clinica Benefit Scale [28]. In the POLO trial, none of the two could be demonstrated for olaparib (S2 File). Hence, In the absence of an increase in quality of life, the utility gains obtained from a PFS benefit without an OS benefit by olaparib are not sufficient to attain cost-effectiveness at internationally accepted thresholds, even when incremental costs would be substantially reduced by lowering the costs for drugs or companion diagnostics.

In conclusion, while olaparib does not meet the criteria for a substantial magnitude of clinical benefit for the treatment of platinum-sensitive metastatic pancreatic cancer, it induces substantial treatment costs. Additionally, the treatment option requires a costly screening of a patient population out of which only few patients may benefit. Olaparib is cleary not a cost-effective treatment option in this patient population at current costs and would have a substantial budget impact. It is important to consider the costs incurred by companion diagnostics when undertaking cost-benefit assessments of targeted therapies in the future.

## Supporting information

**S1 File. Supplementary methods, analysis, tables and figures.**
(DOCX)

**S2 File. ESMO magnitude of clinical benefit scale V1.1.**
(PDF)

## Author Contributions

**Conceptualization:** Tarun Mehra, Judith E. Lupatsch, Roger von Moos, Andreas Wicki, Matthias E. Schwenkglenks.

**Data curation:** Judith E. Lupatsch.

**Formal analysis:** Tarun Mehra, Judith E. Lupatsch, Thibaud Kössler, Andreas Wicki, Matthias E. Schwenkglenks.

**Funding acquisition:** Roger von Moos, Andreas Wicki, Matthias E. Schwenkglenks.

**Investigation:** Tarun Mehra.

**Methodology:** Judith E. Lupatsch, Thibaud Kössler, Andreas Wicki, Matthias E. Schwenkglenks.

**Project administration:** Tarun Mehra, Andreas Wicki.

**Supervision:** Roger von Moos, Andreas Wicki, Matthias E. Schwenkglenks.

**Validation:** Judith E. Lupatsch.

**Writing – original draft:** Tarun Mehra, Judith E. Lupatsch, Thibaud Kössler, Konstantin Dedes, Alexander Reinhard Siebenhüner, Roger von Moos, Andreas Wicki, Matthias E. Schwenkglenks.

**Writing – review & editing:** Tarun Mehra, Judith E. Lupatsch, Thibaud Kössler, Konstantin Dedes, Alexander Reinhard Siebenhüner, Roger von Moos, Andreas Wicki, Matthias E. Schwenkglenks.

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
