## [Decision Letter · Decision Letter 0]

21 Feb 2024

PONE-D-23-40742Olaparib Not Cost-Effective as Maintenance Therapy for Platinum-Sensitive, BRCA 1/2 Germline-mutated Metastatic Pancreatic CancerPLOS ONE

Dear Dr. Mehra,

Thank you for submitting your manuscript to PLOS ONE. After careful consideration, we feel that it has merit but does not fully meet PLOS ONE’s publication criteria as it currently stands. Therefore, we invite you to submit a revised version of the manuscript that addresses the points raised during the review process.

We look forward to receiving your revised manuscript.

Kind regards,

Qi Chen, PhD

Academic Editor

PLOS ONE

Journal Requirements:

2. For studies reporting research involving human participants, PLOS ONE requires authors to confirm that this specific study was reviewed and approved by an institutional review board (ethics committee) before the study began. Please provide the specific name of the ethics committee/IRB that approved your study, or explain why you did not seek approval in this case.

"This study was partially funded by Swiss Study Group for Clinical Oncological Research (SAKK, Schweizerische Arbeitsgemeinschaft für Klinische Krebsforschung)."

"The author(s) received no specific funding for this work"

6. Thank you for stating the following financial disclosure: 

"This study was partially funded by Swiss Study Group for Clinical Oncological Research (SAKK, Schweizerische Arbeitsgemeinschaft für Klinische Krebsforschung)."

7. Thank you for stating in your Funding Statement: 

"This study was partially funded by Swiss Study Group for Clinical Oncological Research (SAKK, Schweizerische Arbeitsgemeinschaft für Klinische Krebsforschung)."

8. Thank you for stating the following in the Competing Interests section: 

"ave read the journal's policy and the authors of this manuscript have the following competing interests:

Roger von Moos: consultancy fees from Astra Zeneca, Eili Lilly, Gilead Science, GlaxoSmithKline, Merck, MSD, Novartis, Pierre Fabre, Pharmamar, Sanofi and Vifor. Travel grants from Pierre Fabre, Takeda"

Reviewers' comments:

Reviewer's Responses to Questions

**Comments to the Author**

1. Is the manuscript technically sound, and do the data support the conclusions?

Reviewer #1: Yes

2. Has the statistical analysis been performed appropriately and rigorously? 

Reviewer #1: Yes

3. Have the authors made all data underlying the findings in their manuscript fully available?

Reviewer #1: Yes

4. Is the manuscript presented in an intelligible fashion and written in standard English?

Reviewer #1: Yes

5. Review Comments to the Author

Reviewer #1: Ref. No: PONE-D-23-40742

I have reviewed the manuscript and found the following areas to be improved in the revised version.

Comment 1: Provide some latest literature in the introduction related to the study.

Comment 2: Mentioned the novelty of the proposed study and gave future recommendations.

Comment 3: The authors should report the mathematical equations of the proposal highlighting all identities.

Comment 4: there are many typos, grammar, and spelling mistakes authors should read the file thoroughly.

Comment 5: It is recommended to provide a Table having all the notations and abbreviations.

6. PLOS authors have the option to publish the peer review history of their article (what does this mean?). If published, this will include your full peer review and any attached files.

Reviewer #1: **Yes: **Zameer Abbas

---

## [Author Response · Author response to Decision Letter 0]

12 Mar 2024

Dear Mr Qi

Many thanks for giving us the opportunity to substantially improve our manuscript. 

We reviewed and amended our manuscript to comply with the journal’s style (title, headings, author affiliations, tables, figures and references). 

We have an ethics statement in the Methods section. As our study uses published or open-source data, an approval from an ethical review board was not mandatory, in accordance with Swiss regulations. Therefore, it was also not necessary to retrieve specific, individual, per-patient informed consent. 

We removed the funding statement from the Acknowledgement section. 

We amended the funding statement, which now reads: “This study was partially funded by the Swiss Study Group for Clinical Cancer Research (SAKK, Schweizerische Arbeitsgemeinschaft für Klinische Krebsforschung). The funder had no role in study design, data collection and analysis, decision to publish, or preparation of the manuscript. There was no additional external funding received for this study.”

We amended the financial disclosure statement to include the Statement: “This study was partially funded by Swiss Study Group for Clinical Cancer Research (SAKK, Schweizerische Arbeitsgemeinschaft für Klinische Krebsforschung)." We also amended the Competing Interests section with the sentences: “All authors have read the journal's policy and the authors of this manuscript have the following competing interests.” and “This does not alter our adherence to PLOS ONE policies on sharing data and materials.”

We have added captions for our Supporting Information files. 

We haven further taken care to respond thoroughly to the Reviewers comments: 

Reviewer 1

We thank the Reviewer for taking time to read and review our manuscript. The valuable comments help us to substantially improving its quality. 

Comment 1: Provide some latest literature in the introduction related to the study.

We conducted a new thorough review of the literature and amended the introduction and discussion. In particular, we now reference the studies of Fashami et al. [1] and Ahn et al. [2]

Comment 2: Mentioned the novelty of the proposed study and gave future recommendations.

We especially thank the Reviewer for pointing out the possibility of highlighting the novelty of our study. We amended the Discussion to this regard. In Strengths and limitations, we added the following sentences to the Strengths and Limitations: “To our knowledge, our study is the first to consider the screening cost of the entire potentially eligibly population in a cost-effectiveness analysis of olaparib.” and “Furthermore, to our knowledge, our study is the first to highlight the substantial financial burden of companion diagnostic testing on the budget impact of introducing olaparib in metastatic pancreatic cancer; a finding which should be considered when evaluating other targeted therapies.”

We give a related recommendation in the conclusion at the end of the Discussion: “Olaparib is cleary not a cost-effective treatment option in this patient population at current costs and would have a substantial budget impact. It is important to consider the costs incurred by companion diagnostics when undertaking cost-benefit assessments of targeted therapies in the future.”

Comment 3: The authors should report the mathematical equations of the proposal highlighting all identities.

We have added the respective equations in the supplement with a reference in the manuscript, where the partitioned survival model is mentioned for the first time. The new section in the Methods of the Supplementary File 1 now reads: 

“A partitioned survival model (PSM) with N states calculates the probability patients being in various health states at a specific time during treatment with a particular therapy. State membership is determined by survival curves that do not overlap; in the case of an N-state model, it necessitates N-1 survival curves.

The cumulative survival function is

 Sn(t) 

It describes the probability of a patient's survival to health state n or to a state with a lower index beyond a specified time t. The likelihood of a patient being in health state 1 is represented by S1(t). Membership in health states 2,…,n −1 is determined by the difference between Sn(t) and Sn−1(t). The final health state N (when patients are dead) is 1−SN−1(t) [3].“

Comment 4: there are many typos, grammar, and spelling mistakes authors should read the file thoroughly.

We conducted a thorough revision of spelling and grammar. 

Comment 5: It is recommended to provide a Table having all the notations and abbreviations.

We are thankful for this suggestion. The additional table has been included as Supplementary Table S1. 

With many thanks and kind regards, on behalf of the co-authors

Tarun Mehra, M.D. 

References

1. Mirzayeh Fashami F, Levine M, Xie F, Blackhouse G, Tarride J-E. Olaparib versus Placebo in Maintenance Treatment of Germline BRCA-Mutated Metastatic Pancreatic Cancer: A Cost–Utility Analysis from the Canadian Public Payer’s Perspective. Current Oncology. 2023;30(5):4688-99. PubMed PMID: doi:10.3390/curroncol30050354.

2. Ahn ER, Rothe M, Mangat PK, Garrett-Mayer E, Calfa CJ, Alva AS, et al. Olaparib in Patients With Pancreatic Cancer With BRCA1/2 Mutations: Results From the Targeted Agent and Profiling Utilization Registry Study. JCO Precision Oncology. 2024;(8):e2300240. doi: 10.1200/po.23.00240. PubMed PMID: 38354330.

3. Network TCRA. Partitioned survival models. 2024.

---

## [Editor Report · Decision Letter 1]

14 Mar 2024

Olaparib not cost-effective as maintenance therapy for platinum-sensitive, BRCA1/2 germline-mutated metastatic pancreatic cancer

PONE-D-23-40742R1

Dear Dr. Mehra,

We’re pleased to inform you that your manuscript has been judged scientifically suitable for publication and will be formally accepted for publication once it meets all outstanding technical requirements.

Kind regards,

Qi Chen, PhD

Academic Editor

PLOS ONE
---

## [Editor Report · Acceptance letter]

26 Mar 2024

PONE-D-23-40742R1 

PLOS ONE

Dear Dr. Mehra, 

I'm pleased to inform you that your manuscript has been deemed suitable for publication in PLOS ONE. Congratulations! Your manuscript is now being handed over to our production team.

Kind regards, 

on behalf of

Dr. Qi Chen 

Academic Editor

PLOS ONE